# A Novel Immunofluorescence Assay for the Rapid Serological Detection of SARS-CoV-2 Infection

**DOI:** 10.3390/v13050747

**Published:** 2021-04-24

**Authors:** Dung Nguyen, Donal Skelly, Niluka Goonawardane

**Affiliations:** 1Nuffield Department of Medicine, University of Oxford, Oxford OX1 3SY, UK; dung.nguyen@ndm.ox.ac.uk (D.N.); donal.skelly@ndcn.ox.ac.uk (D.S.); 2Nuffield Department of Clinical Neurosciences, University of Oxford, Oxford OX1 2JD, UK

**Keywords:** SARS-CoV-2, COVID-19, immunofluorescence assay, variants

## Abstract

As of April 2021, the COVID-19 pandemic has swept through 213 countries and infected more than 132 million individuals globally, posing an unprecedented threat to human health. There are currently no specific antiviral treatments for COVID-19 and vaccination programmes, whilst promising, remain in their infancy. A key to restricting the pandemic is the ability to minimize human–human transmission and to predict the infection status of the population in the face of emerging SARS-CoV-2 variants. Success in this area is dependent on the rapid detection of COVID-19 positive individuals with current/previous SARS-CoV-2 infection status. In this regard, the ability to detect antibodies directed against the SARS-CoV-Spike protein in patient sera represents a powerful biomarker for confirmation of infection. Here, we report the design of a proof-of-concept cell–based fluorescent serology assay (termed C19-S-I-IFA) to detect SARS-CoV-2 infection. The assay is based on the capture of IgG antibodies in the serum of COVID-19-positive patients using cells exogenously expressing SARS-CoV-2-Spike and their subsequent fluorescent detection. We validate the assay in 30 blood samples collected in Oxford, UK, in 2020 during the height of the pandemic. Importantly, the assay can be modified to express emerging Spike-variants to permit assessments of the cross-reactivity of patient sera to emerging SARS-CoV-2 strains.

## 1. Introduction

In December 2019, severe acute respiratory syndrome coronavirus 2 (SARS-CoV-2) emerged in Wuhan, China, causing coronavirus disease 2019 (COVID-19), a highly contagious respiratory illness [1,2]. Since its initial identification in patients with severe pneumonia in Wuhan, infections have rapidly spread across the globe, and the disease was declared a pandemic by the World Health Organization (WHO) on 11 March 2020. As of April 2021, there are 132,004,796 confirmed COVID-19 cases, and total deaths have reached over 2.9 million globally. Current estimates suggest that 568,777 individuals have died from COVID-19 in the United States alone, and more than 126,836 deaths have occurred in the UK. Airborne infection remains the major route of transmission [3,4], and age, male gender, smoking, hypertension, diabetes and cardiovascular disease have been identified as risk factors for severe infections [5,6,7]. For as yet unknown reasons, paediatric patients are at a lower risk of severe COVID-19 disease, accounting for ~90% of all diagnosed asymptomatic or mild cases [8,9,10]. 

Despite the extreme measures taken to restrain the pandemic, the transmission of SARS-COV-2 has continued at an alarming rate [1,11,12,13,14]. A number of European countries are now experiencing a third wave of COVID-19 infections, including Germany, France and Poland. In late 2020 to early 2021, new variants of SARS-CoV-2 with increased transmissibility emerged in the UK (linage B.1.1.7: Variant of Concern 202012/01) and several other countries, including South Africa (linage B.1.351, 501Y.V2) and Brazil (P.1 20J/501Y.V3 variants) [15,16,17,18]. However, their pathogenicity and ability to escape from pre-existing or vaccine-induced antibody-induced immunity still requires investigation. 

In the absence of effective anti-viral therapies for COVID-19, the prevention of human-to-human transmission requires extensive testing, quarantine and contact tracing [3,19]. Although daily rates of new infections in many European countries have declined, regional variability remains an issue. As an example, the reproduction number (R) for SARS-CoV-2 in the North East and Yorkshire, UK, ranges from 0.8 to 1 (as of 5 April 2021) compared to 0.7–1 in the East of England (https://www.gov.uk/guidance/the-r-number-in-the-uk; accessed on 5 April 2021). However, given the low numbers of cases and high degree of variability in regional transmission, these estimates lack the robustness required to inform policy decisions. In addition, the immunization status of individuals to future SARS-CoV-2 variants remains largely unknown, despite the successful implementation of mass vaccination campaigns.

Whilst predictions of the COVID-19 outcome are complex, the identification of individuals previously infected with SARS-CoV-2 can provide knowledge on the degree of immunisation and the individual risk of future infections to emerging variants. In this context, current clinical diagnostic tools (including PCR-based methods) fail to detect immune status [20,21], revealing only those currently infected with the virus. Serological tests can detect IgG, IgA, or IgM anti-SARS-CoV-2 in blood samples and are key to disease surveillance [22,23,24]. Previous studies have confirmed the presence of anti-SARS-CoV-2 IgG/IgM in clinically confirmed COVID-19 cases, even in situations when RT-PCR results were negative [25]. The number of serological tests to detect antibodies against COVID-19 have rapidly increased since the start of the pandemic. These now include enzyme-linked immunosorbent assays (ELISA) [26,27], lateral flow immunoassays (LFIAs), and chemiluminescent immunoassays (CLIAs) [12]. However, commercially available and validated tests would be challenging to upscale if new SARS-CoV-2 variants emerge that can escape patient SARS-CoV-2 antibodies [24,28].

In this study, we report the development of a proof-of-concept cell-based fluorescent serological assay validated for the detection of SARS-CoV-2 status. Using the assay, samples from blood donors collected in Oxford, UK, in March 2020 were assessed for current or previous SARS-CoV-2 infection. We confirm the effectiveness of the immunofluorescence detection method and highlight its applicability for large-scale serological surveillance. Importantly, the assay can be easily modified to express spike protein mutants, permitting predictions of the cross-reactivity of patient sera to emerging SARS-CoV-2 variants, thereby fully informing the immunization status of patients.

## 2. Methods

### 2.1. Patient Samples 

Plasma samples (*n* = 30) were collected in March 2020 during the COVID-19 pandemic in Oxford (UK). Participants had unknown SARS-CoV-2 infection status and were recruited using the Oxford Translational Gastrointestinal Unit GI Biobank Study ethics, code 16/YH/0247 (REC at Yorkshire and the Humber, Sheffield). As negative controls, 100 blood donor samples were collected from the Scottish National Blood Transfusion Service (SNBTS) anonymous archive from September 2018 to December 2019 (IRAS Project No. 18005), prior to the first reports of SARS-CoV-2 in Wuhan. SNBTS blood donors provided informed consent to virological testing. Donations were made under the SNBTS Blood Establishment Authorisation. The study was approved by the SNBTS Research and Sample Governance Committee. Excluded samples included those positive for HIV-1, HCV, HBV or syphilis. 

As positive controls, seven samples from contact-traced individuals were compared (PCR-confirmed as SARS-CoV-2 infected). All infected individuals had asymptomatic COVID-19 and were recruited through the ISARIC WHO Clinical Characterisation Protocol UK (CCP-UK) at the discharge plus 28 day time-point. Samples were heat inactivated prior to serological analysis through incubation at 56 °C for 30 min. 

### 2.2. SARS-CoV-2 Expression Constructs 

Full-length synthetic codon optimised SARS-CoV-2 spike protein (Accession number: YP_009724390.1: Appendix A) was cloned into pcDNA3.1 (+) for mammalian expression under a CMV promoter. pTK-Ren (Promega, Southampton, UK) expressing Renilla luciferase (RLuc) was linearized with *Xba*I and co-transfected as an expression control. DNA quantity and quality were confirmed through Qubit fluorometric quantitation (ThermoFisher Scientific, Waltham, MA, USA). 

### 2.3. Production of the SARS-CoV-2 Spike A549 Cell System 

A549 cells were maintained in DMEM (Life Technologies, Warrington, UK) supplemented with 10% heat-inactivated FCS (Gibco), 100 U/mL penicillin and 100 µg/mL streptomycin (Life Technologies, Warrington, UK). Cells were seeded at 1.5 × 10^5^ cells per well in 96-well plates (*n* = 6; *n* = 3 for IFA and *n* = 3 for Luciferase assay) and transfected with SARS-CoV-2-spike-pcDNA3.1 (+) plasmid (50 ng/well) and 10 ng of pTK-Ren using Lipofectamine 2000 transfection reagent (ThermoFisher Scientific, Waltham, MA, USA). Transfected SARS-CoV-2 spike cells were selected in G418 (Sigma-Aldrich, Steinheim, Germany: 1 mg/mL) and passaged (1:5) every 3 days in 0.5 mg/mL G418 for the generation of stable cell lines. 

### 2.4. SARS-CoV-2-Spike IgG Immunofluorescence Assays (C19-S-I-IFA) 

Serial 2-fold dilutions of each serum sample (1:8 to 1:128 dilutions) were added to A549-SARS-CoV-2-spike cells in 96-well plates for 1 h. Cells were then washed in PBS, fixed in acetone at −20 °C for 15 min, and blocked in PBS-T containing 5% (wt/vol) BSA prior to incubation with anti-human (Hu)-IgG-FITC antibodies (Merck, Darmstadt, Germany; F3512; 1:00 dilution) for 1 h to label bound SARS-CoV-2 spike IgG. Nuclei were counterstained with DAPI (Life Technology, Warrington, UK). Microplates were analysed using a Leica DMIRE2 microscope and Q capture pro 7 software. 

For confocal analysis, A549-SARS-CoV-2-spike cells (5 × 10 ^5^) cells were seeded into 35 mm glass-bottomed culture dishes (MatTek Corporation, MA, USA) and incubated with serum samples at a dilution of 1:40. Cells were then fixed, blocked and labelled with anti-Hu-IgG-FITC antibodies. Confocal images were acquired on a Zeiss LSM880 upright microscope with Airyscan or using an automated Evos FL Auto 2 microscopic system. 

### 2.5. Image Analysis

Post-acquisition analysis was performed using Zen (version 2015 black edition 2.3; Zeiss) or Fiji (version 1.49, software 54). Cells were scored for fluorescence intensity to indicate the presence of IgG against the SARs-CoV2-spike. Test plates included the following controls: (i) PCR-confirmed samples of known COVID-19 status (1/8 and 1:16 dilutions; Table 1); (ii) COVID-19-positive samples added to untransfected cells to rule out non-specific binding; (iii) untransfected and A549-SARS-CoV-2-spike cells labelled with anti-human-IgG-FITC to ensure virus IgG specific fluorescence.

### 2.6. Luciferase Transfection Control Assays 

Cells were assayed for pTK-Ren activity at 24 h post-transfection to ensure comparable levels of exogenous protein expression in A549 cells. Renilla activity was measured using Dual Luciferase reagent kits (50 μL/well, *n* = 3). Total light emission was monitored using the GloMax multi detection system (Promega, Southampton, UK). 

### 2.7. Fluorescence Intensity Scoring (FIS) 

FIS was performed based on the overall staining intensity and percentage of stained cells. Average staining intensities were assigned a value of 0 (no detectable signal) to 4 (highest detectable signal; ≤76% FITC). Staining patterns were not administered a numerical score. Validation criteria are shown in Table 1.

### 2.8. Statistical Analyses 

Statistical significance was determined in GraphPad Prism using a Student’s *t-*test with Welch’s correction or a One-Way ANOVA with Bonferroni’s correction. *p*-values ≤ 0.05 were deemed significant.

## 3. Results

### 3.1. Validation of the C19-S-I-IFA Assay 

To establish antigen detection, samples were compared from patients negative for SARS-CoV-2 infection (collected prior to the pandemic; *n* = 100) and from PCR-confirmed COVID-19-positive individuals (*n* = 5). A549 cells transfected with SARS-CoV-2 Spike (Appendix A)-pcDNA3.1/pTK-Ren were exposed to 2-fold serial dilutions of each serum sample, with the presence of SARS-CoV2 spike IgG in the serum expected to lead to cell labelling, detected following incubation with FITC-conjugated anti-human-IgG (Figure 1). Microscopy was performed rather than flow cytometry as it permitted the more rapid analysis of high sample numbers. Upon analysis, FITC fluorescence was observed in cells treated with serum from COVID-19-positive patients, but was absent in cells incubated with uninfected control samples, confirming specificity (Figure 2). In FITC positive samples, fluorescent labelling was evident at 1:8 and 1:56 fold dilutions of sera, with weaker staining observed at 1:128 (Figure 2A). Positive samples were further analysed by high-resolution Airyscan confocal microscopy (Figure 2B), showing specific cytoplasmic puncta in cells treated with COVID-19-positive sera. Cells exposed to COVID-19-negative sera showed no staining (Figure 2C-D). Upon quantification, the intensity of fluorescence correlated with the serum concentration (Figure 2E). Importantly, the levels of Rluc were comparable in all transfected cell samples, suggestive of reproducible levels of spike protein expression (Figure 2F). Following the exposure of untransfected cells to COVID-19 sera, no fluorescence was observed, confirming the specificity of SARS-CoV-2 spike labelling (Figure 3). We also validated the ability of the assay to be up-scaled through the generation of stable cell lines expressing the SARS-CoV-2 spike and the verification of fluorescent staining using an automated Evos FL Auto 2 microscope system containing an automated multichannel fluorescence counting function (Appendix A). Taken together, these data confirmed the accuracy of the assay for further high-throughput assessments of SARS-CoV-2 positivity and spike protein antibody reactivity. 

### 3.2. Serological Detection of Serum Samples Collected during the 2020 COVID-19 Pandemic 

We next determined the reactivity of unknown samples to determine COVID-19 status (*n* = 30). Fluorescence intensity (Figure 4A) and the percentage of fluorescent-positive cells (Figure 4B) were assessed in cells exposed to 2-fold serial dilutions of each sample. Only a single donor showed reactivity to anti-SARS-CoV-2 spike, with a fluorescence intensity score of 1 at a dilution factor of 1:16, (lower level, ≤25% FITC detection), suggestive of COVID-19 positivity. The positive sample showed no fluorescent labelling in untransfected cells, again ruling out non-specific binding. All other assayed samples were seronegative, with no detectable levels of fluorescence, suggestive of a COVID-19-negative status. These samples were evaluated in parallel by ELISA and showed an identical outcome (data not shown).

## 4. Discussion

The accurate and timely diagnosis of patients with asymptomatic and symptomatic SARS-CoV-2 infections remains crucial to limiting current SARS-CoV-2 human-to-human transmission. However, standard nucleic acid-based molecular diagnosis tools such as RT-PCR and loop-mediated isothermal amplification (LAMP) are dependent on the presence of a sufficient viral load in the upper respiratory tract of infected patients, and are strongly influenced by sample quality. The structural proteins of SARS-CoV-2 are highly immunogenic and lead to the generation of IgM and IgG antibodies [29]. These proteins can be exploited for the development of serological assays such as the enzyme-linked immunosorbent assay (ELISA), the availability of which still fails to meet the global demand. In-house assays that are more scalable are essential to the effective management of the pandemic. Those that can be designed to predict the protection status of previously infected or immunized individuals to emerging SARS-CoV-2 variants are also crucial if the pandemic is to be truly controlled. 

In this study, we report the design of a fluorescent-based cell culture assay using SARS-CoV-2 spike as an antigen for the specific detection of IgG against SARS-CoV-2. A total of 137 serum samples were analysed. We included pre-COVID-19 serum samples (*n* = 100), and seven control samples from contact-traced individuals PCR-confirmed as SARS-CoV-2 positive for assay validation. Thirty of the samples were collected in Oxford (UK), and were of an unknown COVID-19 status. In agreement with clinical PCR findings, the assay accurately detected the presence of SARS-CoV-2 IgG in all seven COVID-19 confirmed cases. As expected, pre-COVID-19 sera (*n* = 100) were negative for anti-spike IgG (Figure 1). Assay positivity also correlated with the serum IgG concentration (Figure 2D,E). Following its validation, the assay identified a single infected patient in the Oxford cohort (Figure 4), revealing its potential as a COVID-19 diagnostic. Importantly, the assay could easily be modified to increase capacity using stable SARS-CoV-2 cell lines and automated high throughput fluorescent analysis (Appendix A).

Recent variants of SARS-CoV-2, including UK variant B.1.1.7 and South African variant 501Y.V2, show clusters of mutations in the spike region that enhance virus transmissibility. The generation of these variants means that new commercial ELISA spike-based serological kits are required to accurately detect infection status. The assay described here provides a proof-of-concept system by which synthetic codon-optimised SARS-CoV-2 spike can be expressed in cultured mammalian cells, which can be expanded through simple Quikchange mutagenesis for rapid adaptation to emerging SARS-CoV-2 variants. This highlights the utility of the assay to supplement current ELISA assessments as new variants emerge and the inevitable global demand for more rapid serological tests grows. The data provided are an accurate representation of that which occurred from February to March 2020 in the UK. We now plan to test the assay for the new circulating variants using mutant cell lines and new patient serum banks.

## Figures and Tables

**Figure 1 viruses-13-00747-f001:**
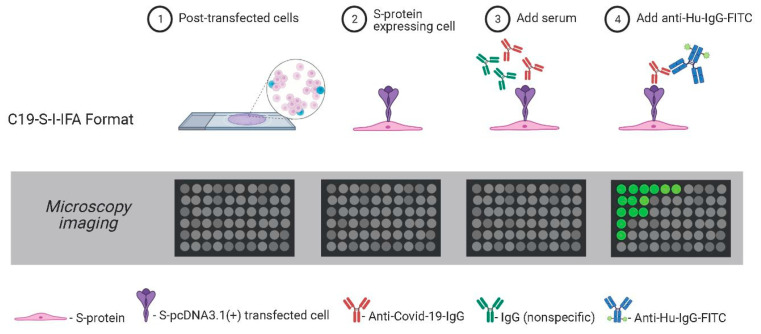
Schematic of the C19-S-I-IFA assay. Serum samples were added to A549-SARS-CoV-2-spike expressing cells. Bound anti-COVID-19-IgG were detected with anti-human-IgG-FITC. Positive cells were indicative of virus-infected sera.

**Figure 2 viruses-13-00747-f002:**
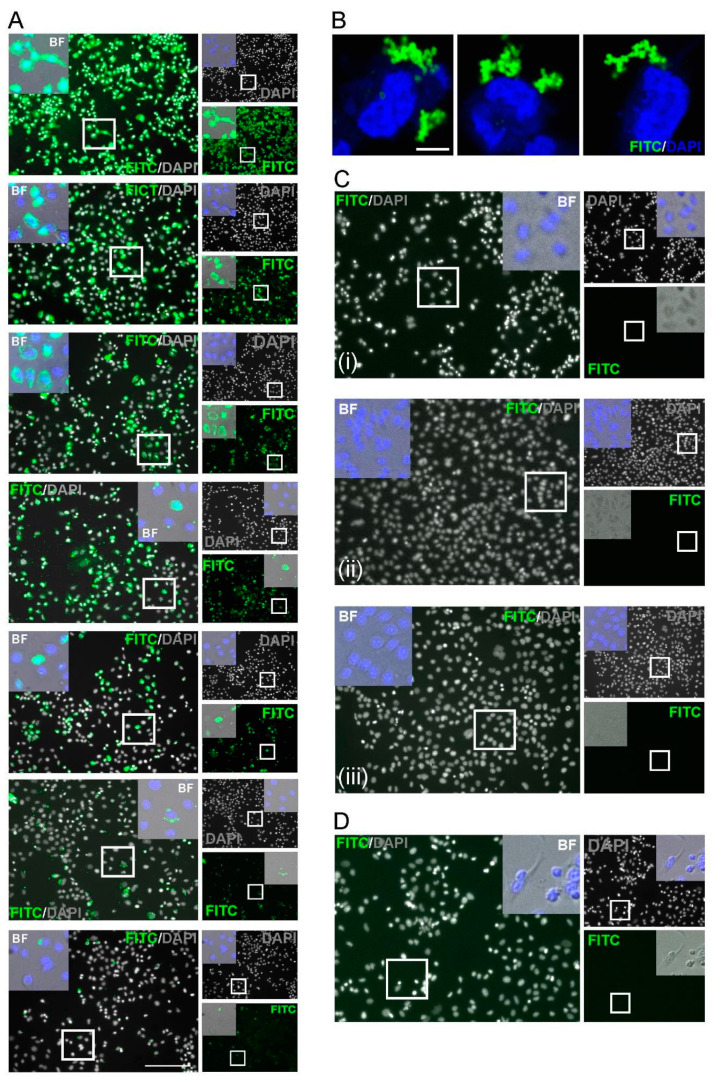
Immunofluorescence characterization of anti-SARS-CoV-2 IgG. (**A**) Indirect staining of A549 cells incubated with serum from a SARS-CoV-2 positive patient and FITC-conjugated antibodies against human IgG. (**B**) Confocal Airyscan images for (**A**) at a dilution of 1:40. (**C**) Negative controls; (i) serum from a PCR confirmed negative donor sample, (ii) spike-expressing cells without serum; (iii) cell-only controls. (**D**) Serum from a pre-SARS-CoV-2 pandemic donor. Nuclei were stained with DAPI (blue). Scale bar: 100 µm and 10 µm. (**E**) Dose–response detection from PCR-confirmed positive donor serum. A two-fold dilution series from 1:8–1:56 was assessed. Bars (green): mean fluorescence intensity; connected circles (blue): % of positive fluorescent cells. Data are from 10 cells per-dilution and were quantified using Fuji. Data are means ± SEM (**F**) pRL-TK Renilla luc (RLuc) was included as an internal transfection control. Bar heights represent the mean of two biological replicates; error bars: SEM.

**Figure 3 viruses-13-00747-f003:**
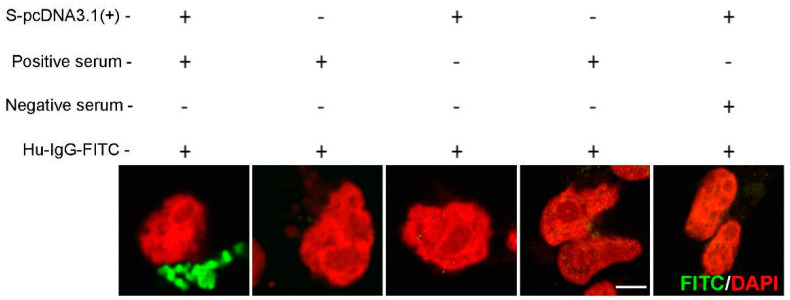
Specificity of spike/anti-SARS-CoV-2 IgG labelling. FITC signals in naive or SARS-CoV-2-spike cells treated with COVID-19-positive or negative serum. Images were analysed using a confocal Airyscan microscopy. FITC: green; cell nuclei: red. Scale bar: 10 µm.

**Figure 4 viruses-13-00747-f004:**
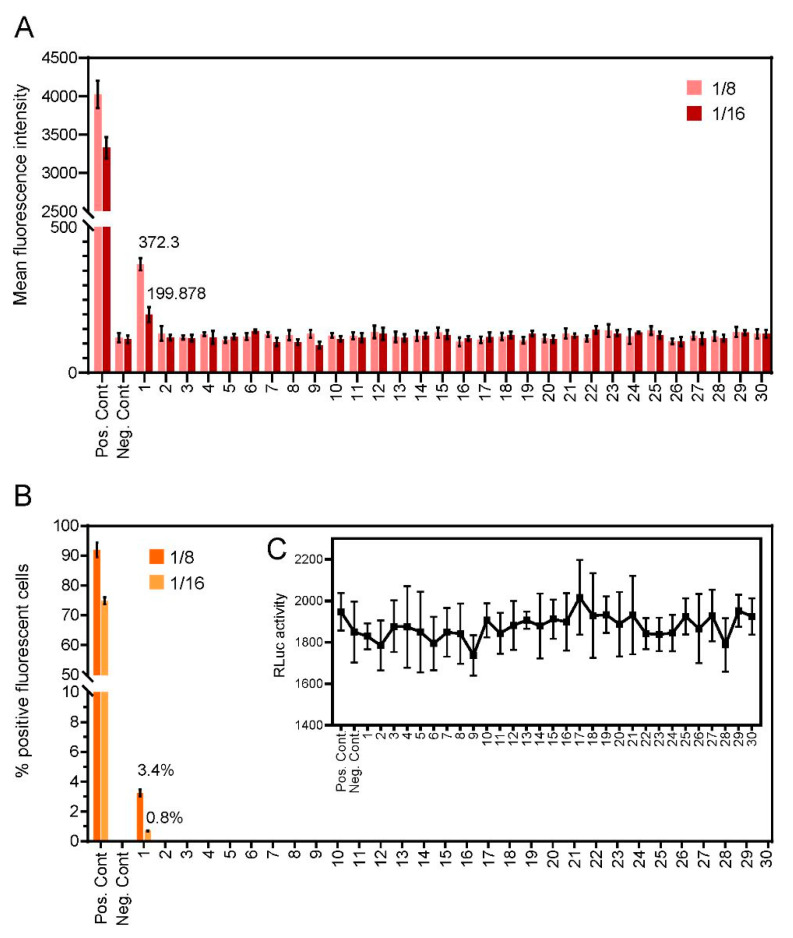
Levels of anti-SARS-CoV-2 IgG antibodies detected through the C19-S-I-IFA assay in the sera from thirty volunteer donors. (**A**) Bar heights represent mean fluorescence intensities for each dilution. (**B**) Percentage of fluorescent positive cells. (**C**) RLuc was used as an internal transfection control. Bar heights represent the mean of two biological replicates: error bars show SEMs.

**Table 1 viruses-13-00747-t001:** Assay validation. Levels of type-specific anti-coronavirus antibodies detected in the human sera of COVID-19-positive or -negative patients.

Method validation and Acceptance Criteria										
Row	Mixture		Antigen dilution #							Cell Controls
			1/8	1/16	1/24	1/32	1/40	1/48	1/56	
# 1	+Ab	+Ag	4	4	3	3	3	2	1	0
≠ 2	+Ab	+Ag	1	0	0	0	0	0	0	0
▪ 3	+Ab	?Ag	1	1	0	0	0	0	0	0
≠ 4	−Ab	+Ag	0	0	0	0	0	0	0	0
◦ 5	−Ab	−Ag	0	0	0	0	0	0	0	0
◊ 6	−Ab	?Ag	0	0	0	0	0	0	0	0

# Fluorescence intensity scores: 0 = no FITC, 1 = ≤ 25%, 2 = ≥ 26%–≤ 50%, 3 = ≥ 49%–≤ 75%, 4 = ≥ 76 %–≤ 100% FITC.; ≠ PCR confirmed positive samples; ▪ Post-pandemic unknown sample (sample ID:1); ◦ Cell control; ◊ Pre-pandemic controls.

## Data Availability

MDPI Research Data Policies at https://www.mdpi.com/ethics (accessed on 24 April 2021).

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
