# Peer review of "A Novel Immunofluorescence Assay for the Rapid Serological Detection of SARS-CoV-2 Infection"

_viruses, 2021, doi:10.3390/v13050747_

Round 1
Reviewer 1 Report
This is an excellent description of a useful way to detect SARS-CoV-2 antibodies in serum using cell lines.
My main concerns are regarding the advantages of this method over others. While I completely agree that ELISAs may be poorly developed and usable at this time, I am not sure how/why this method is a clear advantage over a future usable ELISA (or other method). For instance:
- This requires the maintenance and transfection of a cell line, which has technical limitations that storing ELISA plates do not. Have you managed to develop stable cell lines that could be shipped to labs for testing?
- As I read it, the fluorescence scoring was done manually, which is a significant longer process than an ELISA plate read. Has the fluorescence detection been automated in any way?
- One criticism is the ELISAs might be 'too specific' - but, this method is also based on antibodies binding to specific spike proteins. Has this method been used with spikes from other variants to demonstrate that antibodies to one variant can be detected with cells expressing another variant spike, for example? Or would a lab need to transfect cells with each spike individually?
In saying that, I think this is an interesting method - my main concern is the discussion of how this is an improvement over other methods.
Author Response
Reviewer #1
This is an excellent description of a useful way to detect SARS-CoV-2 antibodies in serum using cell lines. My main concerns are regarding the advantages of this method over others. While I completely agree that ELISAs may be poorly developed and usable at this time, I am not sure how/why this method is a clear advantage over a future usable ELISA (or other method). For instance:
Comment 1: This requires the maintenance and transfection of a cell line, which has technical limitations that storing ELISA plates do not. Have you managed to develop stable cell lines that could be shipped to labs for testing?
Response 1: Thank you for the comment. We have generated stable cell lines expressing the SARS-CoV-2 spike which have been validated for the assay. These have been added as new data in supplementary Figure 1.
Comment 2: As I read it, the fluorescence scoring was done manually, which is a significant longer process than an ELISA plate read. Has the fluorescence detection been automated in any way?
Response 2: We have performed the assay using an automated Evos FL Auto 2 microscopic system, which contains the built-in automated multichannel fluorescence counting function for high throughput analysis. We performed this for the stable cell line discussed above using a standard 96 well-format and included this as new data in supplementary material.
Comment 3: One criticism is the ELISAs might be 'too specific' - but, this method is also based on antibodies binding to specific spike proteins. Has this method been used with spikes from other variants to demonstrate that antibodies to one variant can be detected with cells expressing another variant spike, for example? Or would a lab need to transfect cells with each spike individually?
Response 3: ELISAs for variants would require genetically engineered versions of the spike proteins purified from expressing cells and their addition to the wells of plastic microtiter plates. The assay in this paper would be more rapid as it requires only simple mutagenesis of the spike plasmid and its exogenous expression in mammalian cells for 1-2 days, avoiding purification steps. This provides our assay with flexibility to respond to emerging variants and cost advantages in the face of high numbers of circulating mutant variants. This has been discussed in the revised discussion.
Reviewer 2 Report
I thank the authors for submitting their manuscript to this journal. However, I do not believe that the data is ready for publication in its present form. I recommend the incorporation of additional experiments to validate the IF assay (such as flow) and additional patient samples and controls. Perhaps sera from patients infected with SARS-CoV-2 variants could also be included.
Author Response
Reviewer #2
Comment 1: I thank the authors for submitting their manuscript to this journal. However, I do not believe that the data is ready for publication in its present form. I recommend the incorporation of additional experiments to validate the IF assay (such as flow) and additional patient samples and controls. Perhaps sera from patients infected with SARS-CoV-2 variants could also be included.
Response: We thank you for the comment but respectfully disagree. The assay was validated using 100 control serum samples, 30 volunteers, and numerous controls were shown and discussed to confirm specificity. Microscopy quantification was selected as it is not practically feasible to measure 100 samples by flow cytometry on each assay run. A line describing this has been added to the manuscript. It is also important to note that patient samples are extremely challenging to obtain, with the availability of sequence-confirmed variant samples a near impossibility. We have included two samples to further highlight it as robust, well-controlled and a positive addition to the current tools available to assess COVID-19 infection status.
Reviewer 3 Report
The authors present a well written manuscript and a nice fluorescence method for the detection of antibodies to SARS-CoV-2. The method is clearly described and appear convincing.
The only problem with this manuscript is that in the cohort the authors present is only 1 (one ) positive sample . The authors should enlarge the cohort so that a significant number of positive samples can be presented . Even in Oxford it should not be too difficult colecting sera from SARS-CoV-2antibody positive humans.
Author Response
Reviewer #3
The authors present a well written manuscript and a nice fluorescence method for the detection of antibodies to SARS-CoV-2. The method is clearly described and appear convincing. The only problem with this manuscript is that in the cohort the authors present is only 1 (one) positive sample. The authors should enlarge the cohort so that a significant number of positive samples can be presented. Even in Oxford it should not be too difficult collecting sera from SARS-CoV-2antibody positive humans.
Response: We thank you for the kind comments and recognising that this method is convincing. We used track and trace positive samples for which our findings were in 100% agreement with the infection status of the cohort. This was an accurate representation of that which occurred from February-March 2020 in the UK. To further highlight the accuracy of the assay, we have included two positive control samples in addition to the five PCR-confirmed samples. We plan to test the assay for the new circulating variant using mutant cell lines and more recent patient serum banks, however these are beyond the scope of this paper.
Round 2
Reviewer 2 Report
The authors did not focus their introduction and discussion on SARS-CoV-2 serological assays, comparing the pros and cons with other work (e.g. ELISAs, nanoluciferase biosensors). Many journals have already published in this area and the authors need to highlight any novelty with their assay. I recommend that they remove much of the background on the number of COVID-19 cases, risk factors, disease implications, virus variants, and policy decision making decisions (first three paragraphs), as they do not address any of this with their immunofluorescence images.
Figure 4 remained the same despite authors indicating that seven samples are positive for antibodies against SARS-CoV-2. No new data was added to address reviewer concerns.
The authors transfected the cells with Spike, but did not use their stable cell lines for the screen. The authors did not quantify the cell surface levels of Spike in their transfected cells, or validated that it is specific binding to Spike. Luciferase is not a sufficient control. There are very little validation steps in this paper.
It remains unclear in the text whether microscopy slides are being used. In the schematic (Fig 1) a slide is shown, but is this for validation? Also, are cells stained in microplate format prior to fixation?
It can’t very difficult to obtain VOC samples at the epicentre of the world’s B.1.1.7 outbreak. You must have more than a single positive sample to demonstrate effectiveness of this assay. There is no rational explanation for this not being within the scope of this paper: the variants are widespread and if this assay is incompatible with their detection then it severely lacks utility.
For all samples, display SD instead of SEM. SEM is reserved for biological replicates, and SD is much more relevant in the context of a high throughput technique. Consistency is key, so SD is critical here.
For any high throughput screen, the dynamic range must be calculated and shown. Also include a calculation for the Z-factor, as well as the calculation table for the Z-factor in the supplemental materials. This is an important index that accounts for variability within your samples as well as the detection range between your positive and negative controls .
Display individual data point distributions for all graphs, do not use the summary charts with the error bars alone. Individual points are critical for validating and verifying the power of a high throughput assay.
Our group specializes in high throughput screens. We always see row and/or column effects in biological assays, and this is no exception. It is imperative that you demonstrate %CV in a microplate using a single positive control sample (n=96), and a plate with a negative control sample (n=96). STDev and %CV must be shown for both in the supplementals.
Please check spelling of 'SARS-CoV-2'.
Author Response
We thank you for your kind comments and suggestions. As recommended and in agreement with reviewer 2, we have now highlighted the assay proof-of-concept and highlight its potential to supplement current ELISA assessments should the need outweigh the demand for serological tests in the advent of the emergence of new SARS-CoV-2 variants. All side-by-side comparisons of C19-S-IIFA and standard ELISA assays have been removed or toned down as requested. We have also polished the manuscript and corrected any typos. We thank you for the comments and hope that you now deem the manuscript acceptable for publication.
Reviewer 3 Report
I greater number of positive and negative samples could improve the quality of the manuscript. A publication in its present form however is possible since the overall quality of the manuscript is good.
Author Response
Thank you for your positive comment